# Associations of Elements of Parental Social Integration with Migrant Children’s Vaccination: An Epidemiological Analysis of National Survey Data in China

**DOI:** 10.3390/vaccines9080884

**Published:** 2021-08-10

**Authors:** Shiyu Lin, Zhengyue Jing, Natasha Howard, Tracey Chantler, Jiejie Cheng, Shiya Zhang, Chengchao Zhou, Mei Sun

**Affiliations:** 1Department of Health Policy and Management, School of Public Health, Fudan University, 130 Dong’an Road, Shanghai 200032, China; 20211020128@fudan.edu.cn (S.L.); 18211020050@fudan.edu.cn (J.C.); 20211020100@fudan.edu.cn (S.Z.); 2NHC Key Laboratory of Health Technology Assessment, Fudan University, 130 Dong’an Road, Shanghai 200032, China; 3School of Health Policy and Management, Nanjing Medical University, Nanjing 211166, China; jingzhengyue@mail.sdu.edu.cn; 4Centre for Health Management and Policy, School of Public Health, Cheeloo College of Medicine, Shandong University, Jinan 250012, China; 5Department of Global Health and Development, London School of Hygiene & Tropical Medicine, 15–17 Tavistock Place, London WC1H 9SH, UK; natasha.howard@nus.edu.sg (N.H.); Tracey.Chantler@lshtm.ac.uk (T.C.); 6Saw Swee Hock School of Public Health, National University of Singapore, 12 Science Drive 2, Singapore 117549, Singapore; 7NHC Key Laboratory of Health Economics and Policy Research, Shandong University, 44 Wen-hua-xi Road, Jinan 250012, China

**Keywords:** vaccination, migrant, children, social integration, China

## Abstract

Our study explored the effects of parental social integration on migrant children’s vaccination status in China. Using data obtained from the 2014 China Migrants Dynamic Survey, a total of 4915 participants were included in this study. Social integration was measured by economic, social, cultural, and internal identity. Univariate chi-square testing was used to calculate associations between all variables and migrant children’s vaccination status. Binary logistic regression was employed to calculate the impacts of social integration on migrant children’s vaccination status. In total, 94.7% of migrant children had complete vaccinations for their age. Migrants who had medical insurance, spoke the native language when communicating with locals, lived mainly with locals, and did not perceive discrimination were more likely to have their children completely vaccinated. Social integration was positively associated with migrant children’s vaccination status. Our study indicated that to improve vaccination coverage of migrant children, more policy support for migrant employment and housing, promotion of health services for migrants, and language support in health institutions is needed.

## 1. Introduction

Vaccination is an effective method to prevent infectious diseases, and considered one of the most cost-effective public health services for children [1,2,3,4,5]. Since 2005, the Chinese government has classified vaccines into two categories, with Category 1 including all national-level government-funded and managed vaccines, supplemental vaccines funded by local authorities, and emergency-use vaccines, provided free-of-charge to all children through age 14, and Category 2 (private-sector) including vaccines non-obligatory and paid for by caregivers or persons vaccinated [6]. China has shown impressive achievements in immunization, but emerging challenges due to population migration require attention [7]. Research has demonstrated that migrant children are more vulnerable to missing vaccinations than non-migrant children. A 2010 cross-sectional survey in Delhi, India indicated that migrant children had significantly less chance of being completely immunized than their counterparts [8]. Migration was also identified as an important reason for incomplete vaccination in a qualitative study conducted in Burkina Faso [9]. The 2010 China Census identified 35.8 million migrant children, meaning that 13 out of 100 Chinese children were migrants [10]. With increased urbanization, this number is likely to grow [11]. Studies conducted at city and district levels found that vaccination coverage among migrant children was significantly lower than among non-migrant children [12,13], suggesting this remains a significant public health issue in China.

Many studies have focused on this issue. Factors that may be related to vaccination of migrant children at the individual level, for example, include parents’ education level [3,8,13,14], knowledge and awareness of autonomous immunity [3,12,13], family income [3,14,15,16], healthcare utilization during pregnancy [2,14,16], child age [8,17], birth order [17], and mobility [12,13]. At the institutional level, management practices, potential adverse reactions after vaccination, accessibility of vaccination services [9,17], vaccine supply, and abilities of vaccination personnel [12] were important. In addition, health service policies [14,18,19,20] and mass media affected vaccination information and opinions [3], including influencing the vaccination of migrant children.

Social integration refers to the degree to which migrants adjust to the socioeconomics and culture of a new (i.e., “inflow”) place, and the process of constructing interactive communication and completing a certain degree of psychological adaptation [15,21]. The literature indicates that it can be measured in three dimensions: economic integration, socio-cultural interaction, and internal identity [21,22,23]. It is generally believed that social integration positively affects the health status and health behaviors of migrants, which are closely related to their health outcomes [15]. Few studies have explored the association between social integration and vaccination status in China.

This study sought to analyze the relationship between parental social integration and the vaccination status of migrant children, so as to provide a scientific basis for improving the health status of this population. We had several specific objectives. First, we sought to identify the current status of migrant parents in China with completely vaccinated eldest children. Second, we explored the association between parental social integration and migrant child vaccination status.

## 2. Materials and Methods

### 2.1. Data Sources

Data were derived from the special survey on social integration, part of the 2014 China Migrants Dynamic Survey involving 16,000 migrants in eight provinces using stratified multi-stage probability-proportional-to-size (PPS) sampling. Survey participants were adults aged 15–59 years, living in sampling areas for at least a month without local Hukou registration (a system of household registration used in China). Inclusion criteria for migrant children were those born after June 2007. If participants had more than one child meeting inclusion criteria, the eldest was chosen. Information collected included participants’ basic information, family members, employment and income, housing, and basic health services received.

### 2.2. Dependent Variable

The dependent variable was “completely vaccinated for age”. All participants were asked: “Has your child been completely vaccinated with all Category vaccines (provided by government for free) for his/her age?” as a multiple-choice question with three response categories: yes, no, unclear. We excluded all 323 “unclear” responses, resulting in a total of 4915 children included.

### 2.3. Independent Variables

#### 2.3.1. Socio-Demographics

Socio-demographic characteristics included the following: gender (i.e., male, female); age (i.e., ≤30, >30 years); education (i.e., high school and below, university and above); Hukou (i.e., rural urban); migration period (i.e., ≤5, >5 years); migration range (i.e., between provinces, inside province); eldest child’s gender (i.e., male, female); child age (i.e., ≤3, >3 years); and if the child had received a health examination within the last 12 months (i.e., yes, no, unclear).

#### 2.3.2. Social Integration

We defined social integration in terms of three dimensions: economic integration, socio-cultural interaction, and internal identity. For economic integration variables, we included employment status (i.e., employee, employer, other); total monthly income by quartile, with Quartile 4 the highest; and medical insurance (i.e., yes, no). For socio-cultural interaction, we included communication language (i.e., Mandarin, mother tongue, local language, other); participation in community activities (i.e., yes, no); major neighborhood composition (i.e., migrants, locals, mixed, unclear); and perception of discrimination (i.e., completely disagree, generally disagree, generally agree, completely agree). For internal identity, we included “feeling of being local” (i.e., yes, no) and “willing to settle here” (i.e., yes, no).

### 2.4. Statistical Analysis

We used STATA 14.0 for all calculations. First, we calculated frequencies and chi-squared testing of associations between all variables and eldest child’s complete vaccination status. Second, we conducted binary logistic regression to confirm associations of social integration and eldest child’s complete vaccination status. Finally, we conducted multivariate logistic regression of these associations, to adjust for confounders. Potential confounders adjusted for were employment type, medical insurance, language when communicating locally, neighborhood composition, and perception of discrimination. Statistical significance was set at the 5% level.

## 3. Results

Table 1 shows participant demographics. After excluding participants unsure about their eldest child’s vaccination status, 4652 (94.6%) were included. Approximately 69.8% had been migrants for 5 years or less, and 58.2% migrated outside their province of origin. Chi-square testing showed that completed vaccination status differed significantly by Hukou (*p* = 0.038), child age (*p* = 0.026), and health examination status (*p* = 0.002).

Univariate logistic regression of the association between parental demographic characteristics and the child’s complete vaccination showed that participant gender, age, education, migration period, migration range, child gender, and child health examination in the past year were not associated with completed vaccination. However, parents with rural Hukou had 54% higher odds (OR 1.54; 95%CI 1.02–2.33; *p* = 0.039), and those whose child was over age 3 had 33% higher odds (OR 1.33; 95%CI 1.03–1.70; *p* = 0.026), of their child being completely vaccinated, compared with urban Hukou and younger children, respectively.

Table 2 shows associations between parental social integration and eldest child’s completed vaccination status. Chi-square testing showed employment identity, medical insurance, language when communicating with locals, participation in social activities, major neighborhood composition, perception of discrimination, and the feeling of being local or not differed significantly for those migrants with completely vaccinated children.

Univariate logistic regression of the association between social integration and child vaccination status showed total monthly income and willingness to settle were not associated with complete vaccination. Parents who were employers (OR 2.41; 95%CI 1.33–4.36; *p*-value = 0.004), had medical insurance (OR 1.0), spoke their own language when communicating with locals (OR 1.85; 95%CI 1.09–3.17; *p*-value = 0.024), lived in primarily indigenous neighborhoods (OR 2.18; 95%CI 1.47–3.23; *p*-value < 0.001), did not report discrimination (OR 0.39; 95%CI 0.20–0.74; *p*-value = 0.004), and identified as local (OR 1.0) had greater odds of their eldest child being completely vaccinated than those of their siblings.

Table 3 presents multiple logistic regression analysis of associations between parental social integration exposures and completely vaccinated eldest child, showing both unadjusted and adjusted odds ratios. Adjusted odds ratios showed that migrant parents who were employers (AOR 2.27; 95%CI 1.24–4.16), had medical insurance (AOR 1.0), spoke their mother tongue in daily communication (AOR 1.0), lived in primarily indigenous neighborhoods (AOR 1.75; 95%CI 1.01–3.02), and did not report discrimination (AOR 1.0), had greater odds of their eldest child being completely vaccinated.

Considering that migrant flow status was related to complete vaccination, and the geographic distance of migrant flow in particular was worthy of attention, we conducted further analysis. Table 4 presents a univariate logistic regression analysis of associations between language when communicating with locals and complete vaccination. The result showed that migrant parents who migrated within a province and spoke their mother tongue (OR 1.90; 95%CI 1.04–3.47; *p* = 0.037) or local language (OR 1.20; 95%CI 0.71–2.04) and migrant parents who migrated to an outside province and spoke their mother tongue (OR 3.81; 95%CI 0.53–27.66; *p* = 0.186) with locals had greater odds of their children being completely vaccinated.

## 4. Discussion

The current study found that 94.6 percent of migrant children were completely vaccinated and only 5.4 percent were incompletely vaccinated for their age. This incomplete vaccination rate of the remaining 5.4 percent seems somewhat less significant. However, given that there were 35.81 million migrant children in China according to the 2010 census, nearly 2 million of those migrant children were still not completely vaccinated. Addressing this issue will be of high significance to the goal of achieving a 100 percent vaccination rate among children in China.

Our research showed that economic integration, social and cultural integration, and internal identity of social integration were related to complete vaccination of migrant children in China. In the context of economic integration, employment identity and medical insurance status were positively associated with migrant children’s complete vaccination. Some previous studies also found that the children whose parents had higher-status occupations had a higher likelihood of being completely vaccinated [14,16,24]. One possible explanation is that migrants have improved access to basic and advanced health services if they have better jobs, and thus would be better educated on immunization and have a good understanding of the value of completely vaccinating their children. As for the association between medical insurance and the complete vaccination of migrant children, evidence has shown that migrant children are less likely to see a doctor than their local counterparts, even when they are insured.

In the context of sociocultural integration, major neighborhood composition and language communication with locals significantly affected migrant children’s complete vaccination status. Migrants living with locals had a higher likelihood of being completely vaccinated compared to those living with foreigners. Many studies have explored the effect of neighborhood composition on health status or health behavior [25,26]. Migrants who use their native language to communicate with locals are more likely to have their children completely vaccinated. The association between language and complete vaccination after stratification by migratory range (interprovincial, intra-provincial) supported our speculation that migrants who speak their home language with locals may originate from areas that are not very distant and the differences between the home and local language are within acceptable limits for fluent communication. Close interaction with local neighbors may encourage migrants to learn the local lifestyle, adapt to local life, and even use local health services. Many studies found that the language barrier could significantly affect migrant vaccination status due to linguistic difficulties resulting in poor awareness of vaccination services or of eligibility for vaccines free of charge [27,28].

In the context of internal identity, we found that migrants’ perceptions of discrimination correlated significantly with the complete vaccination status of their children. Few previous studies have explored the association between discrimination and immunization, but a study from Europe also indicated that perceived discrimination was related to poor health outcomes for migrants [29]. We speculate that perceived discrimination may do harm to migrants’ mental health, make their integration into local life more difficult, and then hinder their utilization of health and immunization services.

This study was subject to several limitations. Firstly, our results may not be as accurate as coverage rates for specific vaccines because we selected complete vaccination as the dependent variable in our questionnaire, and this would result in recall bias and selection bias; also, we selected only a few indicators of economic integration, sociocultural integration, and internal identity that may not comprehensively reflect social integration. Secondly, due to a lack of qualitative data, we did not explore how social integration factors as a barrier or facilitator of parents’ behavior in vaccinating their children. Thirdly, we focused only on the factors that improve vaccination rates from the perspective of social integration, and did not discuss other, related factors such as vaccine safety, vaccine hesitance, etc. These limitations could be remedied in follow-up studies.

## 5. Conclusions

Our findings demonstrated that all aspects of parental social integration had an impact on the complete vaccination rate of migrant children. This suggests that China and some other countries, especially developing countries with similar situations, need to show more attention to the social integration status of the migrant population in their relevant health service policies, especially with regard to medical insurance coverage for migrants, the participation of migrants in the local community, and the tolerance and openness of the local community toward migrants. Help migrants achieve better health outcomes with a stronger sense of belonging.

## Figures and Tables

**Table 1 vaccines-09-00884-t001:** Association of socio-demographics of migrant parents in China with complete vaccination of eldest child using univariate logistic regression.

Characteristic	N (%)	Completely Vaccinated (Y/N)	OR	95%CI	*p*
Sample	4915	4652/263	-	-	-
Gender					
Male	3008 (61.2)	2847//161	1.00		0.996
Female	1907 (38.8)	1805/102	1.00	0.78–1.29	
Age (years)					
≤30	2298 (46.8)	2162/136	1.00		0.098
>30	2617 (53.3)	2490/127	1.23	0.96–1.58	
Education					
High school and below	4161 (84.7)	3928/233	1.00		0.069
University and above	754 (15.3)	724/80	1.43	0.97–2.11	
Hukou					
Rural	4215 (85.8)	3978/237	1.00		0.038
Urban	700 (14.2)	674/26	1.54	1.02–2.33	
Migration time (years)					
≤5	3430 (69.8)	3238/192	1.00		0.243
>5	1485 (30.2)	1414/71	1.18	0.89–1.56	
Flow range					
Inter provincial	2859 (58.2)	2706/153			0.998
Intra province	2056 (41.8)	1946/110			
Child gender					
Male	2644 (53.8)	2496/148			0.407
Female	2271 (46.2)	2156/115			
Child age (years)					
≤3	2400 (48.8)	2254/146	1.00		0.026
>3	2515 (51.2)	2398/117	1.33	1.03–1.70	
Child health examination in last 12 months					
Yes	2977 (60.6)	2830/147	1.00		0.002
No	1371 (27.9)	1275/96	0.82	0.64–1.05	
Unclear	567 (11.5)	547/20			

**Table 2 vaccines-09-00884-t002:** Associations of parental social integration among migrants in China with complete vaccination of eldest child using univariate logistic regression.

Characteristic	N (%)	Completely Vaccinated Y/N	OR	95%CI	*p*	Χ2	*p*
Economic integration							
Employment type							
Employees	3041 (61.9)	2873/168	1.0			10.436	0.005
Employers	506 (10.3)	494/12	2.41	1.33–4.36	0.004		
Other	1368 (27.8)	1285/83	0.91	0.69–1.19	0.472		
Total monthly income							
Poorest	1704 (34.7)	1607/97	1.0			1.803	0.614
Less Poor	856 (17.4)	809/47	1.04	0.73–1.49	0.834		
Wealthier	1200 (24.4)	1134/66	1.04	0.75–1.43	0.824		
Wealthiest	1155 (23.5)	1102/53	1.26	0.89–1.77	0.195		
Medical insurance							
Yes	4396 (89.4)	4177/219	1.0			11.202	<0.001
No	519 (10.6)	475/44	0.57	0.40–0.79	<0.001		
Socio-cultural interaction							
Language when communicating with locals							
Mandarin	3385 (68.9)	3208/177	1.0			15.049	0.002
Mother tongue	519 (10.6)	504/15	1.85	1.09–3.17	0.024		
Local language	546 (11.1)	514/32	0.89	0.60–1.31	0.542		
It depends	465 (9.5)	426/39	0.60	0.42–0.86	0.006		
Participate in community activities							
Yes	1740 (35.4)	1663/77	1.0			4.557	0.033
No	3175 (64.6)	2989/186	0.74	0.57–0.98	0.033		
Neighborhood composition							
Migrants	2225 (45.3)	2082/143	1.0			16.534	<0.001
Locals	1014 (20.6)	983/31	2.18	1.47–3.23	<0.001		
Mixed	1450 (29.5)	1376/74	1.28	0.96–1.70	0.097		
Unclear	226 (4.6)	211/15	0.97	0.56–1.68	0.902		
Internal identity							
Experienced discrimination							
Completely disagree	1295 (26.4)	1251/44	1.0			15.359	0.002
Generally disagree	2888 (58.8)	2713/175	0.55	0.39–0.76	<0.001		
Generally agree	576 (11.7)	545/31	0.62	0.39–0.99	0.045		
Completely agree	156 (3.2)	143/13	0.39	0.20–0.74	0.004		
Feel local							
Yes	1105 (22.5)	1065/40	1.0			8.434	0.004
No	3810 (77.5)	3587/223	0.60	0.43–0.85	0.004		
Willing to settle here							
Yes	2638 (53.7)	2511/127	1.0			3.239	0.072
No	2277 (46.3)	2141/136	0.80	0.62–1.02	0.072		

**Table 3 vaccines-09-00884-t003:** Multivariate logistic regression of the relationship between social integration and complete vaccination.

Variables	Unadjusted			Adjusted		
	OR	95%CI	*p*	OR	95%CI	*p*
Economic integration						
Employment identity						
Employees	1.0			1.0		
Employers	2.32	1.27–4.24	0.006	2.27	1.24–4.16	0.008
Self-employed laborer/Else	0.91	0.69–1.20	0.515	0.90	0.68–1.19	0.440
Total monthly income						
Q1	1.0			1.0		
Q2	1.08	0.75–1.55	0.680	1.05	0.73–1.51	0.794
Q3	1.02	0.74–1.42	0.888	0.97	0.70–1.35	0.857
Q4	1.08	0.76–1.54	0.674	1.00	0.69–1.44	0.997
Medical insurance						
Yes	1.0			1.0		
No	0.60	0.43–0.85	0.004	0.61	0.43–0.86	0.005
Socio-cultural interaction						
Language when communicating with locals						
Mandarin Chinese	1.0			1.0		
Home language	1.76	1.02–3.04	0.042	1.75	1.01–3.02	0.045
Local language	0.76	0.51–1.13	0.179	0.74	0.49–1.10	0.132
It depends *	0.59	0.41–0.86	0.006	0.61	0.42–0.89	0.010
Participate in community activities						
Yes	1.0			1.0		
No	0.83	0.63–1.10	0.201	0.86	0.65–1.13	0.277
Major neighborhood composition						
Outsiders	1.0			1.0		
Native	1.88	1.25–2.82	0.002	1.78	1.18–2.68	0.006
Half outsiders and half native	1.27	0.94–1.70	0.117	1.26	0.94–1.70	0.122
Unclear	0.99	0.57–1.72	0.965	1.01	0.57–1.76	0.984
Internal identity						
Discrimination perception						
Completely disagree	1.0			1.0		
Generally disagree	0.63	0.45–0.88	0.008	0.64	0.45–0.90	0.010
Generally agree	0.78	0.48–1.26	0.308	0.80	0.49–1.29	0.352
Completely agree	0.46	0.24–0.88	0.020	0.47	0.24–0.90	0.023
Feel local						
Yes	1.0			1.0		
No	0.73	0.51–1.04	0.080	0.75	0.52–1.07	0.117
Willing to settle here						
Yes	1.0			1.0		
No	0.85	0.65–1.09	0.202	0.87	0.67–1.12	0.275

* If the respondent speaks different languages when communicating on different occasions.

**Table 4 vaccines-09-00884-t004:** Univariate logistic regression of the relationship between social integration and complete vaccination.

Language When Communicating with Locals	Inter Provincial Flow			Intra Province Flow		
	OR	95%CI	*p*	OR	95%CI	*p*
Mandarin Chinese	1.0			1.0		
Home language	3.81	0.53–27.66	0.186	1.90	1.04–3.47	0.037
Local language	0.55	0.29–1.02	0.056	1.20	0.71–2.04	0.502
It depends	0.50	0.28–0.88	0.018	0.73	0.44–1.19	0.206

## Data Availability

The data presented in this study are available on request from the corresponding author.

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
