# Peer review of "Associations of Elements of Parental Social Integration with Migrant Children’s Vaccination: An Epidemiological Analysis of National Survey Data in China"

_vaccines, 2021, doi:10.3390/vaccines9080884_

Round 1
Reviewer 1 Report
The authors have selected an important topic for research, given the often controversial reasons for not receiving the COVID or other recommended vaccines, and as especially related to migrant children. Their results show that the overwhelming number of migrant children ie 94.7% did receive the recommended vaccines, thus making the study of the remaining 5.3% of less significance, also given the complex nature of the individual parameters and interrelationships. These issues could be better addressed in the Discussion.
Author Response
Reviewer 1
Point 1: Their results show that the overwhelming number of migrant children ie 94.7%
did receive the recommended vaccines, thus making the study of the remaining 5.3%
of less significance, also given the complex nature of the individual parameters and
interrelationships. These issues could be better addressed in the Discussion.
Response: Thank you for your kind words and your suggestions. We have
provided more information and revised the discussion. The added content now
reads as follows:
The current study found 94.6 percent of migrant children were completely
vaccinated, and only 5.4 percent of migrant children were incompletely
vaccinated for their age. This incomplete vaccination rate of remaining 5.4
percent seems like to be somewhat less significant. However, given there were
35.81 million migrant children in China according to the 2010 census, there
are still nearly two million migrant children are not completely vaccinated. To
address this issue is of high significance to achieve the goal of 100 percent
vaccination rate among the children in China.
Our research shows that economic integration, social and cultural
integration and internal identity of social integration are related to the
complete vaccination of migrant children in China. In the aspect of economic
integration, employment identity and medical insurance status are positively
associated with migrant children's complete vaccination. Some previous
studies also found that the children whose parents have higher-status
occupations had a higher likelihood of being completely vaccinate [14,16,24].
It may be explained that migrants can get more formal and better health
services if they have a better job, so they would be better educated on
immunization and have a good sense of complete vaccination of their children.
As for the association between medical insurance and complete vaccination of
migrant children, evidence has shown that migrant children are less likely to
see a doctor than their local counterparts, even when they are insured.
Regarding social cultural integration, major neighborhood composition and
language communication with the locals significantly affect the migrant
children’s complete vaccination. Migrants living with the locals have higher
likelihood of being completely vaccinated compared to those living with
foreigners. Many studies had explored the effect of neighborhood composition
on health status or health behavior [25,26]. Migrants who use their native
language to communicate with the locals are more likely to have their children
get complete vaccination. The association between language and complete
vaccination after stratification by flow range (interprovincial, intra-province)
proves our speculation that migrants who speak home language with the locals
may come from places where aren’t very far and the difference between home
and local language is within acceptable limits for fluent communication. Good
connection with local neighbors may promote migrants to learn local lifestyle,
to adopt local life and even to use local health services. Many studies found
that language barrier could significantly affect the migrant vaccination
because of their unawareness of vaccination services or unawareness of their
access for vaccines free of charge due to linguistic difficulties[27,28].
When comes to internal identity, we found that migrants’ discrimination
perception was significantly related with complete vaccination of their
children. Few previous studies explore the association between discrimination
and immunization, but a study from Europe also indicated that perceived
discrimination is related with poor health outcomes in migrants[29]. We
speculate that perceived discrimination may do harm to migrants’ mental
health, make them integrate into local life more difficultly, and then hinder
their utilization of health and immunization services.
This study is subjected to several limitations. Firstly, when we selected
complete vaccination as dependent variable by questionnaires, recall bias and
selection bias may occurred due to self-reported measures; on the other hand,
it may not be as accurate as coverage rates for specific vaccines. Secondly, we
only selected some indicators of economic integration, social cultural
integration and internal identity, which may not comprehensively reflect social
integration. Thirdly, as the lack of qualitative data, we didn’t explore how
social integration factors been as the barrier and facilitators for the parents’
behavior of vaccinating their child which would be remedied in the follow-up
studies. (Discussion section, lines 197-244, p7-8)
Reviewer 2 Report
This paper reports the findings of a study of the effects of parental social integration on migrant children’s vaccination status in China using sample survey data obtained from 4,915 participants from the 2014 China Migrants Dynamic Survey. For the most part, the analyses of the data appear reasonably well done. Here are a couple of items to attend to in a revision.
On p. 7 of the text, it is stated "Table 4 presents Univariate Logistic regression analysis of associations between language when communicate with locals and complete vaccination." But the title of Table 4 states that the contents of the table are from a multivariate regression. This needs to be clarified. And you need to indicate that the entries of Table 4 are from the multivariate regression results in Table 3 (I presume) but are reported in Table 4 for emphasis. And the text of the paper needs to clearly explain the meaning of the entry "It depends".
Author Response
Reviewer 2
Point 1: On p. 7 of the text, it is stated "Table 4 presents Univariate Logistic
regression analysis of associations between language when communicate with locals
and complete vaccination" But the title of Table 4 states that the contents of the table
are from a multivariate regression. This needs to be clarified.
Response: Thank you for this helpful comment, and we are happy to follow.
In response, we have changed our title of table 4 into "Univariate Logistic
regression of the relationship between social integration and complete
vaccination"
Point 2: And you need to indicate that the entries of Table 4 are from the multivariate
regression results in Table 3 (I presume) but are reported in Table 4 for emphasis.
Response: Many thanks to the reviewer for your careful read. In response, we
have added the explanation of the association between Table 3 and Table 4.
The added content now reads as follows:
"Considering that the flow status is related to the complete vaccination,
especially the geographic distance of flow is worthy of attention, we
present the further analysis. Table 4 presents Univariate Logistic
regression analysis of associations between language when
communicating with locals and complete vaccination. The result showed
that migrant parents who flew to the inside province and spoke their
mother tongue (OR 1.90; 95%CI 1.04-3.47; p=0.037) or local language
(OR 1.20;95%CI 0.71-2.04), migrant parents who flew to the outside
province and spoke their mother tongue (OR 3.81; 95%CI 0.53-27.66;
p=0.186) with locals had greater odds of their children being completely
vaccinated." (Results section, line184-191, p7)
Point 3: And the text of the paper needs to clearly explain the meaning of the entry "It
depends".
Response: Thank you very much for your valuable comments. We have
added the annotation below table 3 to explain "It depends". According to
the supplementary instruction of questionnaire, "It depends" means that if
the respondent speaks different languages when communicating on
different occasions. The adjusted content now reads as follows: (Results
section, line181-183, p7)
Table 3. Multivariate Logistic regression of the relationship between social
integration and complete vaccination
Variables
Unadjusted Adjusted
OR 95%CI P OR 95%CI P
Economic integration
Employment identity
Employees 1.0 1.0
Employers 2.32 1.27-4.24 0.006 2.27 1.24-4.16 0.008
Self-employed laborer/
Else
0.91 0.69-1.20 0.515 0.90 0.68-1.19 0.440
Total monthly income
Q1 1.0 1.0
Q2 1.08 0.75-1.55 0.680 1.05 0.73-1.51 0.794
Q3 1.02 0.74-1.42 0.888 0.97 0.70-1.35 0.857
Q4 1.08 0.76-1.54 0.674 1.00 0.69-1.44 0.997
Medical insurance
Yes 1.0 1.0
No 0.60 0.43-0.85 0.004 0.61 0.43-0.86 0.005
Social cultural
interaction
Language when communicate with locals
Mandarin Chinese 1.0 1.0
Home language 1.76 1.02-3.04 0.042 1.75 1.01-3.02 0.045
Local language 0.76 0.51-1.13 0.179 0.74 0.49-1.10 0.132
It depends* 0.59 0.41-0.86 0.006 0.61 0.42-0.89 0.010
Participate activities
Yes 1.0 1.0
No 0.83 0.63-1.10 0.201 0.86 0.65-1.13 0.277
Major neighborhood
composition
Outsiders 1.0 1.0
Native 1.88 1.25-2.82 0.002 1.78 1.18-2.68 0.006
Half outsiders and
half native
1.27 0.94-1.70 0.117 1.26 0.94-1.70 0.122
Unclear 0.99 0.57-1.72 0.965 1.01 0.57-1.76 0.984
Internal identity
Discrimination
perception
Completely disagree 1.0 1.0
Generally disagree 0.63 0.45-0.88 0.008 0.64 0.45-0.90 0.010
Generally agree 0.78 0.48-1.26 0.308 0.80 0.49-1.29 0.352
Completely agree 0.46 0.24-0.88 0.020 0.47 0.24-0.90 0.023
Feel local member or
not
Yes 1.0 1.0
No 0.73 0.51-1.04 0.080 0.75 0.52-1.07 0.117
Willing to settle in
Yes 1.0 1.0
No 0.85 0.65-1.09 0.202 0.87 0.67-1.12 0.275
* If the respondent speaks different languages when communicating on different occasions
Round 2
Reviewer 1 Report
The authors have responded to the suggestions to include a response in the Discussion section. However, it seems unrealistic to expect that vaccination rates can be improved from 94.7% to 100%, even though the remaining unvaccinated numbers of almost 2 million seems worthy of further pursuit. Also not included that would have been of interest and importance are issues of concerns regarding vaccine safety or other reasons for those individuals not receiving or wanting the vaccine. It would be helpful to include this lack of information in the Discussion section.
Reviewer 2 Report
The revisions to this manuscript have been responsive to the previous review, and I have no further comments for revision.
Author Response
Thanks for your patient read and check!